# Reducing manipulations in control simulation experiment based on instability vectors with Lorenz-63 model

Mao Ouyang[1], Keita Tokuda[2, 3], and Shunji Kotsuki[1, 4, 5]

[1]Center for Environmental Remote Sensing (CEReS), Chiba University, 1–33 Yayoi, Inage, Chiba 263–8522, Japan
[2]The Faculty of Health Data Science, Juntendo University, 6–8–1 Hinode, Urayasu, Chiba, 279–0013, Japan
[3]Graduate School of Medicine, Juntendo University, 2–1–1 Hongo, Bunkyo, Tokyo, 113–8421, Japan
[4]Institute for Advanced Academic Research (IAAR), Chiba University, 1–33 Yayoi, Inage, Chiba 263–8522, Japan
[5]PRESTO, Japan Science and Technology Agency, 4–1–8 Honcho, Kawaguchi, Saitama 332–0012, Japan

**Correspondence:** Mao Ouyang (ouyang.mao@chiba-u.jp), Shunji Kotsuki (shunji.kotsuki@chiba-u.jp)

**Abstract.** Controlling weather is an outstanding and pioneering challenge for researchers around the world, due to the chaotic features of the complex atmosphere. A control simulation experiment (CSE) on the Lorenz-63 model, which consists of positive and negative regimes represented by the states of variable $x$, demonstrated that the variables can be controlled to stay in the target regime by adding perturbations with a constant magnitude to an independent model run (Miyoshi and Sun, 2022). The current study tries to reduce the input manipulation of CSE, including the total control times and magnitudes of perturbations, by investigating how controls affect the instability of systems. For that purpose, we first explore the instability properties of Lorenz-63 models without and under control. Experiments show that the maximum growth rate of singular vector (SV) reduces when the variable $x$ was controlled into the target regime. Subsequently, this research proposes to update the magnitude of perturbations adaptively based on the maximum growth rate of SV; consequently, the times to control will also change. The proposed method successfully reduces around 40% of total control times, and around 20% of total magnitudes of perturbations, compared to the case with a constant magnitude. Results of this research suggest that investigating the impacts of control on instability would be beneficial for designing methods to control the complex atmosphere with feasible manipulations.

## 1 Introduction

The ability to control the future based on past and present data is of interest in geophysics, e.g., reducing extreme and rare events, controlling climate changes, and changing cyclone tracks (Lucarini et al., 2016). Miyoshi and Sun (2022) proposed a control simulation experiment (CSE) to change the future by applying perturbations to the independent model run. Their experiments successfully controlled the state variable $x$ in the positive regime of the Lorenz-63 model (Lorenz, 1963; Miyoshi and Sun, 2022). Sun et al. (2022) conducted CSEs on the 40 variables Lorenz-96 models (Lorenz, 1996), results of which demonstrated that CSE can be employed to control the occurrences of extreme events. The goal of CSE is to control the real-world weather (Miyoshi and Sun, 2022), whereas, only preliminary investigations were reported in small-scale dynamic models (Miyoshi and Sun, 2022; Sun et al., 2022). It was demonstrated that the trajectory can be controlled by CSE in ideal models, however, the fundamental principles behind the mechanism of controlling the weather remain obscure. Specifically,

reducing the control times and the magnitudes of the perturbations are two key challenges for a successful implementation of CSEs to complex dynamic systems, which were not thoroughly examined yet. Here, we investigate two types of vectors, namely, bred vector (BV) and singular vector (SV), which are frequently used to explore the instability properties of chaotic models, to try to find feasible manipulations for CSE in the Lorenz-63 model.

Breeding method was proposed to estimate dynamic forecast errors in atmospheric models (Toth and Kalnay, 1993, 1997). BV represents the nonlinear initial error growth within short- and medium-range periods by minimal computation effort. Zhang et al. (2015) demonstrated that the Lorenz-63 model has two different bred vectors, which would converge to one when a small random noise was added to the perturbations at each breeding cycle (Corazza et al., 2003). Singular vector (SV) was defined as the fastest growing perturbations through singular value decompositions of an operator, e.g., tangent linear model (TLM) (Diaconescu and Laprise, 2012). The initial perturbations in ensemble prediction system at the European Centre for Medium-Range Weather Forecasts (ECMWF) were constructed by SVs, due to which produced dispersive ensembles with the most unstable directions (Palmer, 2019). Kim and Jung (2009) identified the regions sensitive to small perturbations in a tropical cyclone by SV. The growth rates of BV and SV were reported to be able to predict the regime changes in the Lorenz-63 model (Evans et al., 2004; Norwood et al., 2013). The calculation of BV and SV could be independent of the infinite time trajectory, which meets the essential of CSE (Miyoshi and Sun, 2022), thus, we examined their properties.

Present study investigates the impacts of control in the Lorenz-63 model on BV and SV and discusses the approach of introducing these vectors to determine feasible manipulations in CSE. We first review the approach of CSE briefly (Miyoshi and Sun, 2022), and describe the methods for calculating BV and SV. Then, two trajectories of the Lorenz-63 model: one without control, and the other one with control activated at a certain time, are calculated. We compare the BVs and the SVs of the two trajectories at the initial stage of activating control and during a long time period. Based on the features of vectors, we will discuss possible approaches to adaptively determine the manipulations in CSE.

## 2 Method

### 2.1 Lorenz-63 model

The model used in this study is Lorenz-63 model, given by:

$$\dot{x} = \sigma(y - x), \tag{1}$$
$$\dot{y} = x(\rho - z) - y, \tag{2}$$
$$\dot{z} = xy - \beta z, \tag{3}$$

where the standard parameters $\alpha = 10$, $\rho = 28$, and $\beta = 8/3$ are selected for chaotic behaviour with two regimes i.e., the famous butterfly pattern (Lorenz, 1963). The initial condition is chosen as $(x, y, z) = (8.20747939, 10.0860429, 23.86324441)$ following Miyoshi and Sun (2022). Nature run (NR) is obtained by integrating the Lorenz-63 model for 208,000 time steps using the fourth-order Runge-Kutta scheme with a time step increment $(\mathrm{d}t)$ of 0.01. Hereafter in this paper, we use the number of these

discretised time steps as the time parameters. We save the NR for 208,000 time steps because it could provide sufficient long reference data for evaluating the data assimilation results and provide many starting points for investigating the characteristics of CSEs under various conditions. This trajectory is named as $x_n$, where subscript $n$ represents NR without control.

## 2.2 Control simulation experiment

The CSE is an output feedback control method that determines manipulations based on outputs (or signals) from the system. In Miyoshi and Sun (2022)'s framework, the output is observation data generated from NR. To estimate accurate analysis, the CSE needs to employ sequential data assimilation cycles. This study implements the ensemble Kalman Filter (EnKF) (Bishop et al., 2001; Houtekamer and Zhang, 2016) following Miyoshi and Sun (2022). Three initial forecast ensembles are obtained by adding random Gaussian noise $r \sim \mathcal{N}(\overrightarrow{5.0}, 1.0\mathbf{E})$ to NR (Kalnay et al., 2007), where $\mathbf{E}$ is the identity matrix, $\overrightarrow{5.0}$ denotes the vector $(5, 5, 5)^{\mathrm{T}}$, and $\mathcal{N}(\overrightarrow{5.0}, \Sigma)$ denotes the multivariate normal distribution applied throughout the paper and defined as bellow:

$$\mathcal{N}(\overrightarrow{5.0}, \Sigma) \equiv \frac{1}{(2\pi)^{\frac{3}{2}}|\Sigma|^{\frac{1}{2}}} \exp(-\frac{1}{2}(r - \overrightarrow{5.0})^{\mathrm{T}} \Sigma^{-1}(r - \overrightarrow{5.0})). \tag{4}$$

Observations are generated every $T_a = 8$ time steps by adding Gaussian noise $\mathcal{N}(\overrightarrow{0.0}, 2.0\mathbf{E})$ to NR, and are assimilated in the data assimilation cycle. Multiplicative inflation of the ETKF is manually tuned to be 1.04, which is consistent with previous studies (Kalnay et al., 2007). The first 8,000 time steps, corresponding to 1,000 data assimilation cycles, are discarded to ensure that the root mean square error (RMSE) of the mean of ensemble forecast and NR is around 0.30 (Yang et al., 2012). The purpose of the ensemble data assimilation cycle is to obtain ensemble forecasts for determining manipulations in CSE.

The goal of CSE is to control state variable $x$ to stay in the positive regime by adding perturbations to NR. The process designed by Miyoshi and Sun (2022) is reviewed as follows:

1. Perform a data assimilation based on observations at time $t$.

2. Employ an ensemble forecast for $T$ time steps from $t$ to $t + T$.

3. If at least one ensemble member changes the regime, control (step 4) will be activated; otherwise, data assimilation at time $t + T_a$ will be performed for the next cycle (step 1).

4. Perturbations are defined as the differences between the ensemble forecasts with and without regime changes. If all three ensemble members show regime change, former initial ensembles will be forecasted for extended time steps to identify at least one ensemble member showing no regime change from $t$ to $t + T$. The perturbations normalized by constant Euclidean norm $D$ are added to NR at every time step from $t + 1$ to $t + T_a - 1$. The new NR, i.e., NR with adding perturbations, is used to generate observations for subsequent data assimilation cycle (step 1).

The forecast time period $T$ and Euclidean norm of perturbations $D$ are tunable parameters in CSE. The ratio of successful control was high when $T = 300$ time steps and $D = 0.05$ (Miyoshi and Sun, 2022), thus, we adopt these parameters in this study. The trajectory under control by CSE is named as $x_c$, where the subscript $c$ represents NR under control.

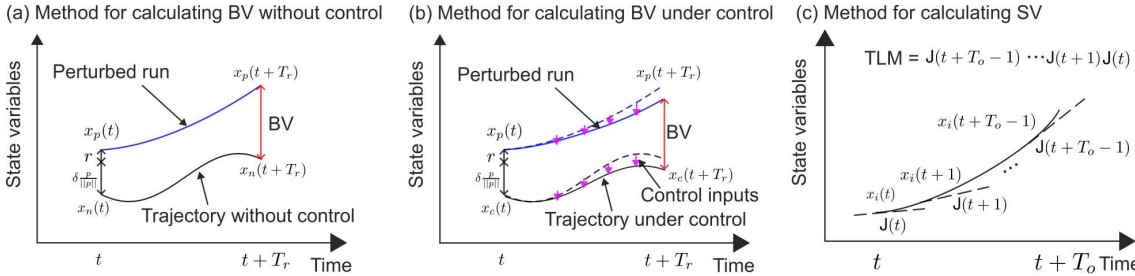

**Figure 1.** Methods for calculating (a) BV without control, (b) BV under control, and (c) SV in general case. Please refer to the text for the meaning of each parameter.

## 2.3 Bred vector

Of the two types of vectors, BV is the easier and faster to be computed. We need to define the trajectory of interest $x_i$, which is $x_n$ or $x_c$ in this study. Figures 1 (a) and (b) show the methods for calculating BV without and under control. At the beginning of breeding cycle at time $t$, a perturbation $p$, scaled to size $\delta$, and a Gaussian noise $r \sim \mathcal{N}(\vec{0}, 0.1\mathbf{E})$ (Corazza et al., 2003), are added to the trajectory $x_i(t)$.

The initial condition for the perturbed trajectory $x_p(t)$, at time $t$, will be:

$$x_p(t) = x_i(t) + \delta \frac{p}{||p||} + r, \tag{5}$$

where $||p||$ is the Euclidean norm of $p$. The Lorenz-63 model is then used to integrate the perturbed trajectory forward starting from $x_p(t)$ for both trajectories without control $x_n$ and under control $x_c$. For the trajectory under control $x_c$, the influence of control inputs in CSE accumulates over multiple time steps. To examine the influence of the perturbation $(\delta \frac{p}{||p||} + r)$, on the systems without and under control, we add the same control inputs (the purple arrows in Fig. 1 (b)) to the perturbed run as the independent run $x_c$ for calculating the BV. At the end of the rescaling interval $T_r$, BV can be obtained by subtracting $x_i(t+T_r)$ from the perturbed trajectory,

$$\mathrm{BV}(t+T_r) = x_p(t+T_r) - x_i(t+T_r). \tag{6}$$

The process is repeated from Eq. (5) with $p = \mathrm{BV}(t+T_r)$ as the perturbation for the next breeding cycle. The growth rate of BV is calculated by $\frac{1}{T_r} \ln \frac{||\mathrm{BV}||}{\delta}$, where $||\mathrm{BV}||$ is the Euclidean norm of BV. The main parameters of BV are the perturbation size $\delta$, and rescaling interval $T_r$, which represent the effects of linear and nonlinear disturbances on error growth. We set $\delta$ equals to 1.0 and $T_r$ equals to 8 time steps following Evans et al. (2004). The growth rate of BV is calculated once every $T_r$ = 8 steps when the norm of BV is normalized. To obtain an instantaneous BV growth rate at every time step, we used parallel BVs, each normalized successively at different time points along the trajectory.

 **2.4 Singular vector**

The vectors which maximize the growth rate of perturbations for a chosen norm and optimization time interval ($T_o$) can be represented by SV. The process of finding SV for a given state ($x_n$ or $x_c$) starts with the Jacobian matrix (Diaconescu and Laprise, 2012), which of Lorenz-63 model at a given state ($x(t), y(t), z(t)$), is given by

$$\mathbf{J}(t) = \begin{bmatrix} -\sigma & \sigma & 0 \\ \rho - z(t) & -1 & -x(t) \\ y(t) & x(t) & -\beta \end{bmatrix}. \tag{7}$$

For simplicity, we set $T_o$ equals to 1 time step, i.e., 0.01 time units. The SV can be calculated through the singular value decomposition of Jacobian matrix (7) as:

$$\mathbf{J}(t) = \mathbf{U}\mathbf{S}\mathbf{V}^{\mathbf{T}}, \tag{8}$$

where $\mathbf{U}$ and $\mathbf{V}$ are orthonormal matrices (Press et al., 1992), $\mathbf{V}^{\mathbf{T}}$ denotes the conjugate transpose of $\mathbf{V}$, $\mathbf{S} = \mathrm{diag}(s_{11}, s_{22}, s_{33})$ is a diagonal matrix with descending non-negative singular values. We calculate the first column of $\mathbf{V}$, which is the initial leading SV, corresponding to the fastest growing vector from time $t$ to $t+1$. Euclidean norm is employed to compute the growth rate of SV, which is given by $\ln s_{11}$.

We show the general case of calculating SV for dynamic models in Fig. 1 (c). The singular value decomposition is conducted in the tangent linear model (TLM) by the production of Jacobian matrices. In this study, the Jacobian matrices for both trajectories without and under control ($x_n$ and $x_c$) are calculated by Eq. 7, assuming that the manipulations applied in the CSE can be regarded as an external force that would not affect the linear propagation of perturbations.

## 3 Results

### 3.1 Control simulation experiment

We first conduct the experiments to confirm that CSE can control the state variable $x$ to the target regime. Experimental results without control, i.e., trajectory with orange lines ($x_n$), and under control, i.e., trajectory with magenta lines ($x_c$), are shown in Fig. 2. Spin-up states refers to the initial period taken before the control is activated where the variables might suffer from transit effects (Lorenz, 1996), which are shown as blue lines in Fig. 2. Observation of Figs. 2 (b) and (d) demonstrates that CSE can successfully generate observations and control the state variable $x$ in the positive regime.

We calculate BV and SV of the two trajectories, $x_n$ and $x_c$, based on the methods described in Section 2. To investigate the influence of control on the vectors, we focus on the instantaneous changes of vectors when control is activated and the changes of vectors during a long time period, which will be shown subsequently.

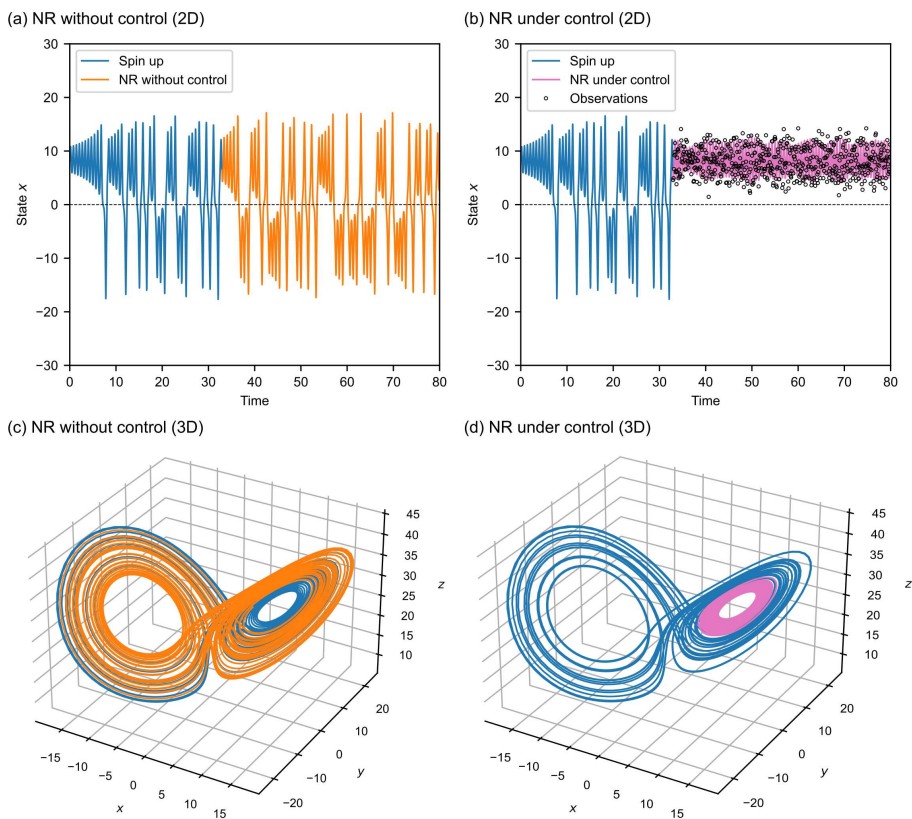

**Figure 2.** State variables of the Lorenz 63 model. (a) State variable $x$ in 2D plane without control. (b) After control, state variable $x$ stays in the positive regime. The empty circles represent the observations generated by CSE. (c) Lorenz's butterfly attractor from NR, i.e., no control ($x_n$). (d) Trajectory under control ($x_c$). Blue line represents the spin-up states. Orange and magenta lines represent the NR without and under control, respectively. Initial control is activated at the time of 32.89.

## 3.2 Influence of starting point on CSE

Lorenz (1963) noted that when the state variable $x$ shows large values, the trajectory tends to change regimes. This suggests that if the starting point of the control is near the extreme value of $x$, we may be more difficult to control the state variables in the target regime. Here, we conduct 400 CSE experiments with different starting points randomly sampled in between the time unit of 50 and 150. When the state value $x$ of the starting points is positive, the CSE experiments are performed for 8,000 time steps, i.e., 1,000 data assimilation cycles. Figure 3 shows the relationship between the number of successful and failed CSE experiments and the state variable $x$ when the initial control is activated. If the control is activated at state variable $x$ in the range of 15–20, the failed probability is quite high. For the successful controls, the initial controls are occurred in the range from 0 to 15.

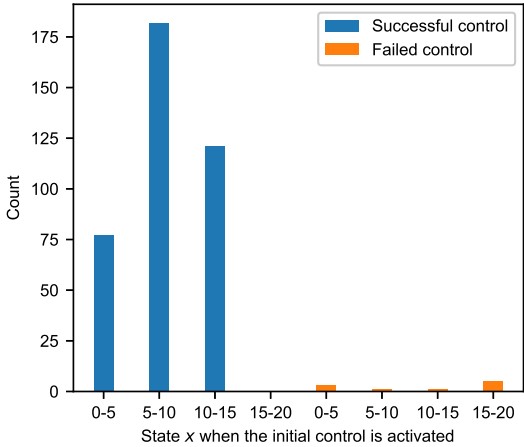

**Figure 3.** The numbers of successful and failed CSE experiments with the state variable $x$ when the initial control is activated.

## 3.3 Influence of control at the initial stage

Figure 4 shows the vector changes without and under control at the initial stage of experiments. After discarding the first 8,000 time steps, we start the CSE, the initial control is activated at the time of 32.89. Figure 4 (a) shows the time to start control and the different trajectories without and under control by orange lines $x_n$, and magenta lines $x_c$, respectively. The empty circles are the observations generated by the CSE, which are all located in the target regime. Figures 4 (b) and 4 (c) show changes in BV and SV for experiment without and with control inputs. The lengths of the vectors represent the growth rates of BVs and SVs, which are enlarged by a factor of 20 and 50 for better visualization, respectively. When the control inputs are added to the model, both the direction and magnitude of BV are changed due to that the external forces would change the nonlinear error propagations for $T_r$ time steps, i.e., 8 time steps. The directions and magnitudes of SV are similar for those without and under control (Fig. 4 (c)). This is because that both the NRs without and under control during the selected period do not show the trend to change regimes. Therefore, even though the control inputs are added to the NR under control, the direction and magnitude are similar to that of NR without control.

## 3.4 Influence of control during a long time period

It was reported that the regime changes of the Lorenz-63 model can be predicted by the growth rates of BV (Evans et al., 2004) and SV (Norwood et al., 2013). In the case of BV, the regime change was predicted by the growth rate, i.e., when the growth rate in current cycle exceeds 0.064, next cycle will change to the other regime (Evans et al., 2004). Norwood et al. (2013) reported that the growth rate of SV also implied the regime changes, i.e., when the growth rate of SV exceeds 0.0296 in current cycle, state variable $x$ would change to the other regime in the next cycle.

Examined vectors of the trajectory without control are shown in Figs. 5 (a) and (b) for 2,500 time steps. Table 1 shows the forecast verifications (Jolliffe and Stephenson, 2011) based on the rules of regime changes prediction described above, where

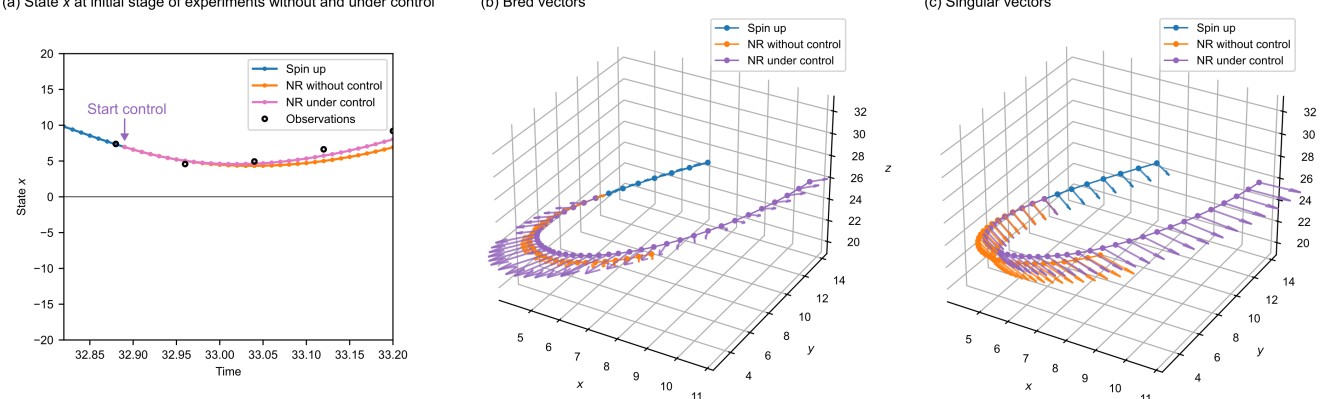

**Figure 4.** Experimental results of changes in two vectors at the initial stage of CSE. (a) State $x$ at the initial stage of experiments without and under control. The empty circles represent the observations generated by CSE. (b) The BVs. The lengths of the vectors represent the growth rate of BV with a magnification of 20. (c) The SVs. The lengths of the vectors represent the growth rate of SV with a magnification of 50. Blue line represents the spin-up states. Orange and magenta lines represent the NR without and under control, respectively. Control is activated at time 32.89.

"hit" means the rule successfully forecasts the observation; "miss" means that regime change is observed but not forecasted by the rule; and "false alarm" means that the rule forecasts a regime change but no regime change occurs; "correct rejection" means that the regime change is neither observed nor forecasted by the rule. The threshold values used here are described in the above paragraph. We verify the forecast for 100 different time series with each verification period equal to 1,000 time units, i.e., 100,000 time steps. The percentages of mean values and the standard deviations are shown in Table 1. Results indicates

that the growth rate of SV shows better performance than BV regarding the regime change prediction.

**Table 1.** Regime change forecast verification of BV and SV. The numbers and those in the bracket represent the percentage of mean and standard deviation to the averaged total number of 100 different time series.

|      | Hits (%)   | Misses (%) | False alarms (%) | Correct rejections (%) |
|------|------------|------------|------------------|------------------------|
| BV   | 33.3 (1.0) | 9.8 (0.7)  | 5.1 (0.6)        | 51.8 (1.3)             |
| SV   | 42.1 (1.4) | 1.0 (0.5)  | 7.7 (0.7)        | 49.2 (1.7)             |

The features of BV and SV of trajectory under control are shown in Figs. 5 (c) and (d). Characteristics of BV and SV are altered by the control. Figure 5 (c) suggests that the growth rate of BV of controlled system shows large values despite the absence of regime changes, which gives us two hints. One is that when external forces are added to Lorenz-63 model, BV may not be able to effectively predict the regime changes. Another one is that the nonlinear error growth of the controlled system

is larger than that of the original system, indicating that CSE might increase the nonlinear features to the chaotic models. To investigate the Lyapunov dimensions through covariant Lyapunov vectors (Norwood et al., 2013; Ginelli et al., 2013, 2007) is

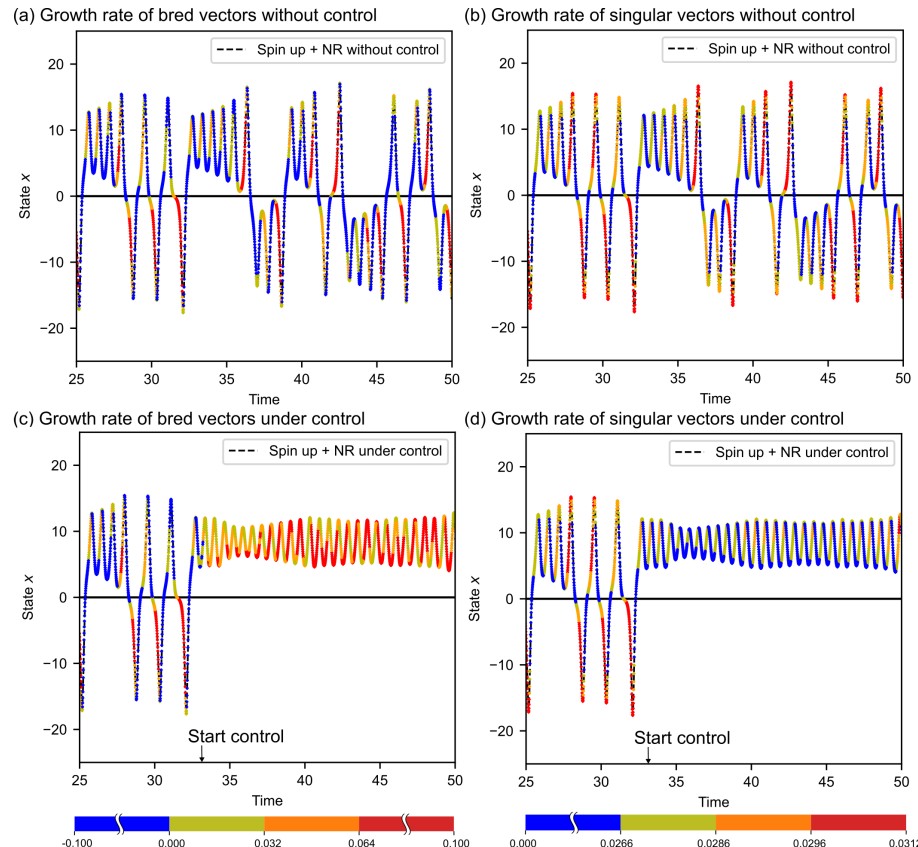

**Figure 5.** Experimental results for long time period. Growth rates of BV (a) without control and (c) under control. Growth rates of SV (b) without control and (d) under control. Control is activated at time 32.41. The black dashed lines are the trajectories of interest, $x_n$ and $x_c$. Coloured stars in (a) and (c) represent the growth rate of BV, with absolute value shown in the colour bar below (d). Coloured stars in (b) and (d) represent the growth rate of SV, with absolute value shown in the colour bar below (e).

our future study for extensively exploring the chaotic characteristics. On the other hand, the growth rate of SV is reduced after control, e.g., maximum growth rate shows 5 % decrease, i.e., from 0.0312 to 0.0296 (Fig. 5 (d)).

To illustrate the robustness of the reduction of the maximum growth rate of SV, we examined the SV of the successful CSEs with different starting points. The relationship between the state variable $x$ when the initial control is activated and the maximum growth rate of SV is shown in Fig. 6. The maximum growth rates of SV are affected by the starting point. Compared with the cases without control, the maximum growth rate of SV under control presents a great reduction for all CSEs with different starting points. The maximum growth rates of SV under control shows a mean value of 0.0296 and a standard deviation of 0.00015. We use the mean value of 0.0296 for our subsequent studies.

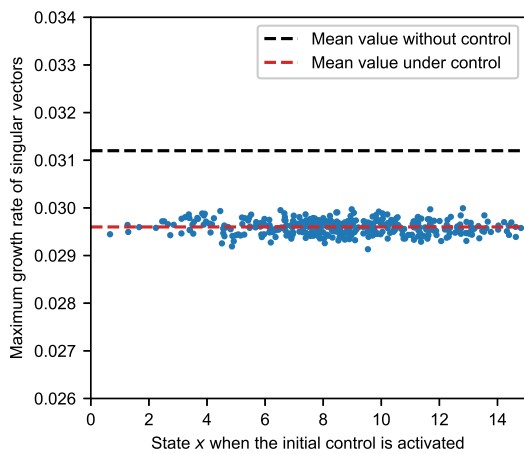

**Figure 6.** Maximum growth rate of SV for the trajectories under control at different starting points.

 ## 4 Discussion

Our results demonstrated that controlling the state variables changed the characteristics of BV and SV. Here, we will discuss a new approach which could determine feasible manipulations, including total control times and magnitudes of perturbations in CSE, through the insights from the investigations on instability vectors.

### 4.1 Introduce growth rate of SV to update magnitude adaptively

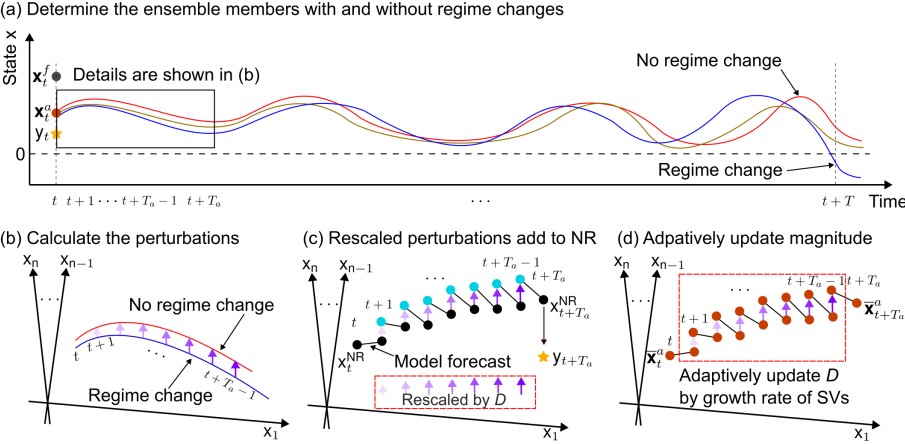

**Figure 7.** Schematic illustrations of the experimental procedures. (a) Determine the ensemble members with and without regime changes. (b) Calculate the perturbations based on the differences of the ensemble forecasts obtained from (a). (c) Rescale the perturbations to magnitude, $D$, and add them to NR. (d) Adaptively update the magnitude $D$ according to the growth rate of SV of analysis mean. Please note that the axes $x_1$, $x_{n-1}$, and $x_n$ in (b), (c), and (d) represent that the perturbations are multidimensional vectors in this study.

Miyoshi and Sun (2022) controlled the state variables by adding constant magnitude of perturbations to NR in their CSE. The schematic diagram of the process related to manipulation in CSE is shown in Figs. 7 (a-c). After performing the data assimilation at time $t$, analysis ensembles are iterated for $T$ time steps to find the ensemble members with and without regime changes (Fig. 7 (a)), the differences of which during $t+1$ and $t+T_a+1$ over 0.08 time units are regarded as perturbations (Fig. 7 (b)). State variables are changed by adding the rescaled perturbations with constant magnitude $D$, to NR from $t+1$ to $t+T_a-1$ (Fig. 7 (c)). Miyoshi and Sun (2022) suggested that the magnitude of perturbation was a sensitive parameter, and needed further investigations.

Investigation of SV showed that when the state $x$ was controlled in positive regime, the maximum growth rate of SV was less than 0.0296 (Section 3.4). We then propose to apply this rule in CSE to update the magnitude of perturbations added to NR adaptively, as shown in Fig. 7 (d). The perturbations (Fig. 7 (b)) are rescaled with initial Euclidean norm $D = 0.010$, and are added to the integration of analysis mean $\overline{\mathbf{x}}_t^a$, which is the mean value of the analysis ensembles after the data assimilation (step 1 of CSE), from $t+1$ to $t+T_a-1$. We calculate the growth rates of SVs for these $T_a-1$ time steps. The magnitude $D$ is determined until the maximum growth rate of SVs was less than 0.0296, otherwise increase $D$ by 0.001. The process will finish after either finding $D$ which meets the requirement, or $D$ reaching the boundary which is assumed as the maximum possible intervention.

We conduct experiments with only one data assimilation cycle to investigate the applicability of growth rate of SV in CSE. When $D$ is subjectively determined, Fig. 8 (a) shows that the cases with too small (0.010) and too large (1.400) magnitudes cannot successfully control the state $x$ in positive regime, whereas, that with magnitude 0.050 can achieve the goal. If $D$ is determined by the growth rate of SV, Fig. 8 (b) notes that the necessary magnitude to control state $x$ in positive regime is only 0.021. This test elaborates that the growth rate of SVs can be applied in CSE to successfully control state variables.

## 4.2 Comparison between constant and adaptive magnitudes

To investigate the influence of adaptive magnitude on the total control times and magnitudes of perturbations, we conduct CSEs with constant magnitude, 0.05, as recommended by Miyoshi and Sun (2022), adaptive magnitude without boundary, and adaptive magnitude with boundary to be 0.05, for 500 time steps. The results are shown in Fig. 9 and Table 2. The total control times reduce from 62.4 % (312 out of 500) to 22.4 % (112 out of 500) if the growth rate of SV was implemented to update the magnitude adaptively. The case with adaptive magnitude and boundary shows around 20 % decrease of total magnitudes of perturbations compared to that with constant magnitude.

We conduct 40 experiments for 8,000 time steps, corresponding to 1,000 data assimilation cycle, to examine the performance of adaptive magnitude and the sensitivity of $D$ for constant magnitude, as shown in Fig. 10. Rate of successful control is calculated as the ratio of cases which NR is controlled in the target regime. Figure 10 (a) shows that rate of successful control for adaptive magnitude was around 97.5 %, suggesting only one case fails out of 40. Total control times by adaptive magnitude without boundary are reduced from those by constant magnitude, when the applied constant magnitudes are smaller than 0.200 (Fig. 10 (b)). Total control times of cases with adaptive $D$ by growth rate of SV and boundary are larger than those with only adaptive $D$ by growth rate of SV (blue and magenta lines in Fig. 10 (b)). Experiment with adaptive magnitude by growth rate

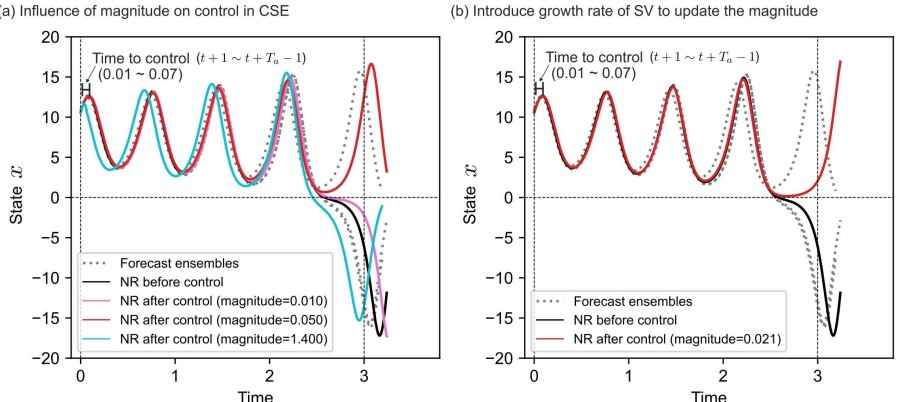

**Figure 8.** Introducing growth rate of SV to control the state variables. (a) Influence of magnitudes on the control in CSE. (b) Introduce growth rate of SV to calculate the necessary magnitudes. The grey dotted lines are the forecast ensembles. Black lines represent the NR before control. Perturbations are added to NR from time 0.01 to 0.07 over 7 time steps. The purple and cyan lines represent the NR after control with constant magnitude of perturbations 0.010 and 1.400, respectively. The red line in (a) represents the NR after control with subjectively determined constant magnitude of perturbation 0.050. The red line in (b) represents the NR after control with magnitude of perturbations determined by growth rate of SV, 0.021.

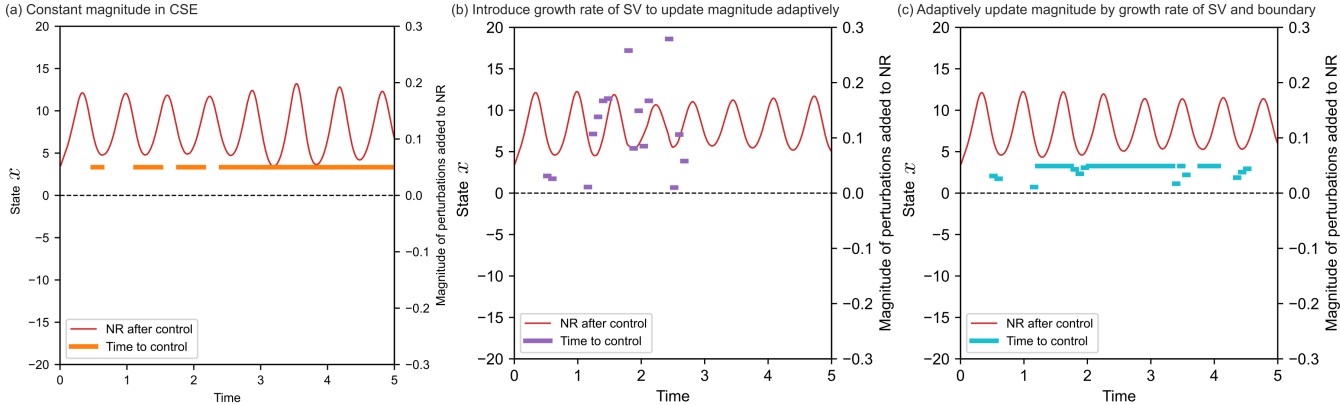

**Figure 9.** Comparison between constant and adaptive magnitudes in CSE for 500 time steps. (a) Constant magnitude in CSE. (b) Introduce the growth rate of SV to update the magnitudes adaptively. (c) Adaptively update the magnitude by growth rate of SV and boundary. The red lines represent the NRs after control. The thick orange, magenta, and blue lines represent the times to activate control by constant magnitude (a), adaptive magnitude without boundary (b), and adaptive magnitude with boundary (c), respectively.

**Table 2.** Comparison between constant and adaptive magnitudes for 500 time steps.

| | The number of control inputs | Total magnitudes |
|---|---|---|
| Constant $D$ | 312 | 15.6 |
| Adaptive $D$ by growth rate of SV | 112 | 12.9 |
| Adaptive $D$ by growth rate of SV and boundary | 259 | 12.4 |

of SV and boundary achieves the minimal total magnitudes of perturbations, around 90.6, among all experiments in this study
(Fig. 10 (c)).

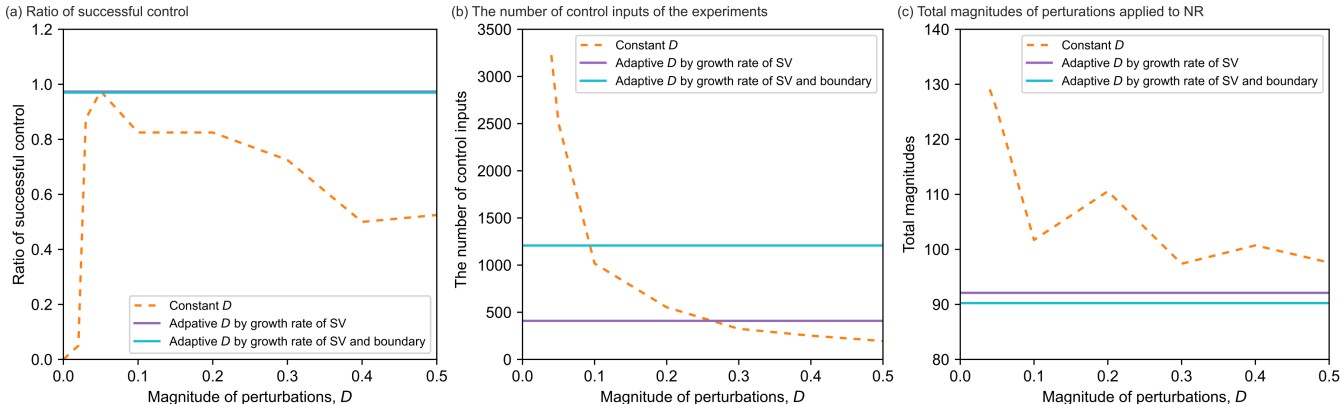

**Figure 10.** Comparison between constant and adaptive magnitudes in terms of (a) Ratio of successful control; (b) Times to control; (c) Total magnitudes. The orange dashed, magenta, and blue lines represent the results of constant $D$, adaptive $D$ by growth rate of SV, and adaptive $D$ by growth rate of SV and boundary, respectively. Please note that the magenta and blues lines are the results with adaptive $D$, which are not dependent on the prescribed $D$ and are shown as horizontal lines in this figure.

## 5    Conclusions

We investigate the impacts of control on two vectors related to instability, i.e., bred vector and singular vector in the Lorenz-63 model. Results demonstrate that the features of these vectors will change if the state variables are controlled to the target regime. Maximum growth rate of singular vectors shows around 5 % decrease after control, and is introduced to calculate
the magnitude of perturbations in control simulation experiment. Accordingly, the manipulation, including total control times and magnitudes of perturbations, is designed to be updated adaptively based on the maximum growth rate of singular vectors. Total control times and magnitudes of perturbations show around 40 % and 20 % reduction, respectively, when updating the magnitude adaptively, which indicates that the manipulations are reduced through the method proposed in this research.

The information of bred vector and singular vector may be possible to be applied to reduce the time of control and magnitude of perturbations for the general cases, include but not limit to Lorenz-63 model and regime changes. Control simulation experiment was designed to control the weather which shows separatrices, which could be defined as the separations between a normal precipitation and a heavy rainstorm, or the separations of typhoon moving east and moving west etc. The error growth near these separatrices might presents different characteristics compared with the that far away from the separatrices, e.g., a great change of the growth rate of singular vector. Based on the information provided by the instability vectors, we may be able to find an effective way for reducing the manipulations in the control simulation experiment.

Control the state variables of the Lorenz-63 model is a simplified scenery of meteorological study. For complex dynamic systems, e.g., full-scale numerical weather prediction model, cautions are necessary when apply the proposed method to control the weather. For example, it is difficult to explicitly obtain the tangent linear model in medium and large scale atmospheric models, and a larger number of ensemble size would cause a higher probability for the ensemble forecasts to the undesired regimes, thus, more studies are necessary on the adaptive manipulation update. Present study throws lights on the introduction of instability vectors to find optimal manipulations for the control simulation experiment. Climate change-induce extreme weather, e.g., typhoon, storm surge, intensive rainfall, was reported to cause catastrophes in recent years (Kotsuki et al., 2019; Ouyang et al., 2021, 2022). Covariant Lyapunov vectors could provide insights into the spatio-temporal instability of chaotic systems (Egolf et al., 2000). Future research will be focused on exploring the chaotic features by covariant Lyapunov vectors (Ginelli et al., 2007; Tokuda et al., 2019, 2021), and implementing these vectors in atmospheric models to control extreme weather for mitigating natural hazards.

*Code availability.* The code that supports the findings of this study is available from the corresponding authors upon reasonable request.

*Data availability.* The authors declare that all data supporting the findings of this study are available within the figures and tables of the paper.

*Author contributions.* MO and SK conceived of the study. SK is the principal investigator. MO conducted the numerical experiments and wrote the manuscript. KT analysed the results. All authors contributed to framing and revising the paper.

*Competing interests.* The authors declare that they have no conflict of interest.

*Acknowledgements.* This study was partly supported by JST PRESTO MJPR1924, the Program for Promoting Research on the Super-computer Fugaku of MEXT JPMXP1020200305, the Japan Society for the Promotion of Science (JSPS) KAKENHI grants JP21H04571

JP22K18821, JP20K19882, JST Moonshot R&D JPMJMS2284, and the IAAR Research Support Program of Chiba University. The authors thank project members of the Moonshot JPMJMS2284 for fruitful discussion.

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
