# Peer review of "Reducing manipulations in control simulation experiment based on instability vectors with Lorenz-63 model"

_Nonlinear Processes in Geophysics, 2023_

## Referee Comment (RC1)

**Comments on the manuscript entitled "Reducing manipulations in control simulation experiment based on instability vectors with Lorenz-63 model"**

2023-02-15

The authors extend the control simulation experiments (CSEs) by Miyoshi and Sun [2022] wherein the state of the Lorenz-63 model is constrained in its positive regime. Growth rates of bred vector (BV) and singular vector (SV) have been assessed to explore the instability properties of the dynamical system in the CSEs. The SV results in reducing the total control times and perturbation magnitude (i.e. Euclidean norm of perturbations) than the BV and constant-magnitude perturbation [Miyoshi and Sun, 2022]. This study suggests substituting perturbations with a constant magnitude [Miyoshi and Sun, 2022] to adaptive values based on the growth rates of the SV in CSEs. The presentation of the current manuscript is concise and easy to follow overall. However, I expect that the authors will address major questions/comments before further decisions for publication. First, the authors should improve the Introduction which will better show the motivations of their study and facilitate the readers to capture its novelty. More importantly, the manuscript is lacking an intensive analysis of the experiments presented therein. Section Results almost focuses on the experiment description rather than the results' interpretation. Details of my general comments can be found below. Some other points are also listed for consideration in the manuscript revision.

1. **Abstract**: The authors have mentioned the BV-based method in the abstract but do not provide it a conclusion. I suggest adding a sentence for comparing its performance to the SV or removing the term 'bred vector' from the abstract.

2. **Introduction**: Reading the Introduction, I feel it is like a brief review of BV and SV. Highlights of CSEs' applications in practice and the pros and cons of the previous proposed approaches would be more appropriate. From these points, the authors could suggest to use BV and SV...

   (a) Lines 16-19: Please split these sentences precisely. Miyoshi and Sun [2022] tested the experiments on the L63 model and Sun et al. [2022] proposed CSEs on the L96 model.

(b) Lines 21-23: There exist any other approaches which can be used to learn the instability properties of chaotic models (e.g. Lyapunov Exponents)? Why should BV and SV be the first candidates to be examined? Refer to Norwood et al. [2013] for general ideas.

3. **Method**:

   (a) Line 48: Nature runs for 208000 time steps: can you give a hint for this number?

   (b) Line 49: "with the width of each time step (dt) equal to 0.01" → "with a time step increment (dt) of 0.01"

   (c) Lines 55-56: Please double-check! Miyoshi and Sun [2022] used EnKF while this study has employed ETKF. Do different data assimilation methods impact the results of CSEs? And ensemble forecasts with a larger ensemble size (3 in this study) would challenge the CSEs as it gives a higher probability for the forecasts to switch between the two L63 regimes?

   (d) Lines 111-112: The growth rate of SV is not formulated precisely?

4. **Results**

   (a) Figure 2: Can you plot the observations as points in the same plot with the time series? I am curious to see their illustration in the CSEs.

   (b) Lines 129-134: Please elaborate the analysis for Figure 3. Four of the five sentences describe the experiment and figure details...

      • Personally, I think the successful rate of CSEs would highly depend on the starting point of the control activation. It would be interesting to verify the sensitivity of CSEs on the starting points (e.g. the state x is between 0-5, between 5-10, 10-15, etc).

      • It is necessary to interpret the different performances of BV and SV in Figures 3b and 3c as they are the core of this study.

   (c) Lines 140-146 (Table 1): The total numbers of hit, miss, and false alarm events are not the same for BV (412) and SV (387). Were the experiments executed on the same trajectory? I suggest to verify the forecast with different time series of L63 and then computing the confidence interval of Hits, Misses, False Alarms, and threat Scores.

   (d) Line 148: "Fig.4d" → "Fig.4c"

   (e) Line 150: "Fig.4e" → "Fig.4d", 0.0312 does not appear on the color bar.

5. **Discussion**: Line 168: "2.96" → "0.0296". It is not convinced that the maximum growth rate of SV (0.0296) is fixed for any starting point in CSEs. The authors could plot the maximum growth rate of SV as a function of the starting point to see whether the maximum growth rate varies or not.

6. For figures, please increase the line width for better visualization.

**References**

T. Miyoshi and Q. Sun. Control simulation experiment with lorenz's butterfly attractor. *Nonlinear Processes in Geophysics*, 29(1):133–139, 2022.

A. Norwood, E. Kalnay, K. Ide, S.-C. Yang, and C. Wolfe. Lyapunov, singular and bred vectors in a multi-scale system: an empirical exploration of vectors related to instabilities. *Journal of Physics A: Mathematical and Theoretical*, 46(25):254021, 2013.

Q. Sun, T. Miyoshi, and S. Richard. Control simulation experiments of extreme events with the lorenz-96 model. *Nonlinear Processes in Geophysics Discussions*, pages 1–18, 2022.

---

## Author Comment (AC1)

**Reviewer #1**

**General comments** — The authors extend the control simulation experiments (CSEs) by Miyoshi and Sun (2022) wherein the state of the Lorenz-63 model is constrained in its positive regime. Growth rates of bred vector (BV) and singular vector (SV) have been assessed to explore the instability properties of the dynamical system in the CSEs. The SV results in reducing the total control times and perturbation magnitude (i.e. Euclidean norm of perturbations) than the BV and constant-magnitude perturbation (Miyoshi & Sun, 2022). This study suggests substituting perturbations with a constant magnitude (Miyoshi & Sun, 2022) to adaptive values based on the growth rates of the SV in CSEs. The presentation of the current manuscript is concise and easy to follow overall. However, I expect that the authors will address major questions/comments before further decisions for publication. First, the authors should improve the Introduction which will better show the motivations of their study and facilitate the readers to capture its novelty. More importantly, the manuscript is lacking an intensive analysis of the experiments presented therein. Section Results almost focuses on the experiment description rather than the results' interpretation. Details of my general comments can be found below. Some other points are also listed for consideration in the manuscript revision.

**Reply**: We thank the reviewer for careful and thorough reading of the manuscript, and for detailed suggestions and comments, which help us to improve the presentation and quality of this contribution.

We have revised the Introduction section: instead of reviewing the BV and SV, we highlighted the goals and challenges of the CSEs according to the suggestions from the reviewer.

Regarding the Results section, we added more discussions and analysis on the results, e.g., influence of starting points on the CSE, and their relationship to the maximum growth rate of SV.

Please see below, in blue, for a point-by-point response to the reviewers' comments and concerns. All page numbers refer to the annotated manuscript with tracked changes.

The revisions were made throughout the manuscript.

**Reviewer comments 1.1 Abstract** — The authors have mentioned the BV-based method in the abstract but do not provide it a conclusion. I suggest adding a sentence for comparing its performance to the SV or removing the term 'bred vector' from the abstract.

**Reply**: As per the suggestions from reviewer, we removed the term 'bred vector' from the abstract. The sentence was revised to be:

"For that purpose, we first explore the instability properties of Lorenz-63 model without and under control. Experiments show that the maximum growth rate of singular vector (SV) reduces when the variable $x$ was controlled into the target regime".

The revision was made in Lines 6–9.

**Reviewer comments 1.2 Introduction** — Reading the Introduction, I feel it is like a brief review of BV and SV. Highlights of CSEs' applications in practice and the pros and cons of the previous proposed approaches would be more appropriate. From these points, the authors could suggest to use BV and SV $\cdots$

**Reply**: We thank you for suggesting the structure of Introduction section. Two articles were published on the applications of CSE, which were added in the revised manuscript. The pros of CSE was that it could successfully control the variables to the target regimes. However, until now, no theoretical proofs on the dynamical chaotic models were shown on 1) when to start control, and 2) how many manipulations are necessary for the successful control, to our best knowledge. To deepen the understanding of CSE, we introduce the instability vectors BV and SV to try to understand the instability impacts of CSE.

"The goal of CSE is to control the real-world weather (Miyoshi & Sun, 2022), whereas, only preliminary investigations were reported in small-scale dynamic models (Miyoshi & Sun, 2022; Sun et al., 2022). It was demonstrated that the trajectory can be controlled by CSE in ideal models, however, the fundamental principles behind the mechanism of controlling the weather remain obscure. Specifically, reducing the control times and the magnitudes of the perturbations are two key challenges for a successful implementation of CSEs to complex dynamic systems, which were not thoroughly examined yet."

The revisions were made in Lines 23–28.

**Reviewer comments 1.2 (a) Introduction** — Lines 16–19: Please split these sentences precisely. Miyoshi and Sun (2022) tested the experiments on the L63 model and Sun et al. (2022) proposed CSEs on the L96 model.

**Reply**: We split the sentences with respect to Lorenz-63 and Lorenz-96 models as following:

"Miyoshi and Sun (2022) proposed a control simulation experiment (CSE) to change the future by applying perturbations to the independent model run. Their experiments successfully controlled the state variable $x$ in the positive regime of the Lorenz-63 model (Lorenz, 1963; Miyoshi & Sun, 2022). Sun et al. (2022) conducted CSEs on the 40 variables Lorenz-96 models (Lorenz, 1996), results of which demonstrated that CSE can be employed to control the occurrences of extreme events."

The revisions were made in Lines 18–22.

**Reviewer comments 1.2 (b) Introduction** — Lines 21–23: There exist any other approaches which can be used to learn the instability properties of chaotic models (e.g. Lyapunov Exponents)? Why should BV and SV be the first candidates to be examined? Refer to Norwood et al. (2013) for general ideas.

**Reply**: We select BV and SV as the first candidates, because 1) they can present the nonlinear and linear features of error growth in the dynamic models; 2) they are independent of the infinite time trajectory. The essential of CSE is that the prediction and consequent

manipulations are blind to the true states (Miyoshi & Sun, 2022), which means that the infinite time trajectory was unknown in the experimental design. Lyapunov vector (LV) can be employed to investigate the instability properties of chaotic models, however, the calculation of LV needs the information of the infinite time trajectory, since it involves backward calculation from the global Lyapunov exponents (Norwood et al., 2013). The manipulations added to the chaotic systems would change the global Lyapunov exponents, which makes the calculation of LV difficult due to the non-autonomous systems. We will explore the influence of CSE on LV in the future publications. The reasons for selecting BV and SV as the first candidates and the plan are revised as follows:

"The calculation of BV and SV could be independent of the infinite time trajectory, which meets the essential of CSE (Miyoshi & Sun, 2022), thus, we examined their properties."

"Covariant Lyapunov vectors could provide insights into the spatio-temporal instability of chaotic systems (Egolf et al., 2000). Future research will be focused on exploring the chaotic features by covariant Lyapunov vectors (Ginelli et al., 2007; Tokuda et al., 2019, 2021), and implementing these vectors in atmospheric models to control extreme weather for mitigating natural hazards."

The revisions were made in Lines 40–41 and Lines 262–264.

**Reviewer comments 1.3 (a) Method** — Line 48: Nature runs for 208000 time steps: can you give a hint for this number?

**Reply**: We run the model without control for 208,000 time steps because it could 1) provide sufficient long time reference data for evaluating data assimilation results, e.g., more than 2,000 analysis cycles after discarding the first 8,000 time steps (Kalnay et al., 2007; Yang et al., 2012); and 2) provide many starting points for investigating the characteristics of CSEs under various conditions. The reason was added as follows:

"We save the NR for 208,000 time steps because it could provide sufficient long reference data for evaluating the data assimilation results and provide many starting points for investigating the characteristics of CSEs under various conditions."

The revisions were made in Lines 58–60.

**Reviewer comments 1.3 (b) Method** — Line 49: "with the width of each time step (dt) equal to 0.01" → "with a time step increment (dt) of 0.01"

**Reply**: Fixed.

The revisions were made in Line 57.

**Reviewer comments 1.3 (c) Method** — Lines 55-56: Please double-check! Miyoshi and Sun (2022) used EnKF while this study has employed ETKF. Do different data assimilation methods impact the results of CSEs? And ensemble forecasts with a larger ensemble size (3 in this study) would challenge the CSEs as it gives a higher probability for the forecasts to switch between the two L63 regimes?

**Reply**: We thank the reviewer for carefully checking the original paper. Houtekamer and Zhang (2016) reviewed that the ensemble Kalman filter (EnKF) includes both stochastic and deterministic filters. Miyoshi and Sun (2022) performed the Observing Systems Simulation Experiment (OSSE) following previous studies (Kalnay et al., 2007; Yang et al., 2012). These studies employed the ensemble transfer Kalman filter (ETKF) for data assimilation (Bishop et al., 2001) on the Lorenz-63 model. To keep consistence with the previous studies, we revised the manuscript to be:

"This study implements the ensemble Kalman Filter (EnKF) (Bishop et al., 2001; Houtekamer & Zhang, 2016) following Miyoshi and Sun (2022)."

The purpose of data assimilation is to accurately estimate the analysis. Hence, if the data assimilation can be properly performed, different data assimilation method would have negligible impacts on the results of CSEs.

We agree with the reviewer's comments that a larger ensemble size would give a higher probability for the forecasts to change Lorenz-63 regimes. More manipulations are necessary for CSEs when a larger number of ensemble forecasts change regimes. These might be challenging the for the practical application of CSE, and we addressed the limitation in the Conclusion section as follows:

"For example, it is difficult to explicitly obtain the tangent linear model in medium and large scale atmospheric models, and a larger number of ensemble size would cause a higher probability for the ensemble forecasts to the undesired regimes, thus, more studies are necessary on the adaptive manipulation update."

The revisions were made in Lines 65–66 and Lines 257–259.

**Reviewer comments 1.3 (d) Method** — Lines 111–112: The growth rate of SV is not formulated precisely?

**Reply**: We revised the method for calculating growth rate of SV to make it concise. The growth rate of SV is given by $\ln s_{11}$, where $s_{11}$ is the first singular value of the singular value decomposition of Jacobian matrix at time $t$.

The revision were made in Lines 128–129.

**Reviewer comments 1.4 (a) Results** — Figure 2: Can you plot the observations as points in the same plot with the time series? I am curious to see their illustration in the CSEs.

**Reply**: We added the observations in the time series under control by CSE, as shown in Fig. 2 (b). Figure 2 (b) demonstrates that the CSE could successfully generate the observations in the target regime.

The revisions were made in Line 143 and Fig. 2.

**Reviewer comments 1.4 (b) Results** — Lines 129-134: Please elaborate the analysis for Figure 3. Four of the five sentences describe the experiment and figure details...

• Personally, I think the successful rate of CSEs would highly depend on the starting point

[Figure]

Figure 2: State variables of the Lorenz 63 model. (a) State variable $x$ in 2D plane without control. (b) After control, state variable $x$ stays in the positive regime. The empty circles represent the observations generated by CSE. (c) Lorenz's butterfly attractor from NR, i.e., no control ($x_n$). (d) Trajectory under control ($x_c$). Blue line represents the spin-up states. Orange and magenta lines represent the NR without and under control, respectively. Initial control is activated at the time of 32.89.

of the control activation. It would be interesting to verify the sensitivity of CSEs on the starting points (e.g. the state x is between 0-5, between 5-10, 10-15, etc).
- It is necessary to interpret the different performances of BV and SV in Figures 3b and 3c as they are the core of this study.

**Reply**: We elaborated the analysis for Fig. 3 (of first submission manuscript), and conducted additional experiments on the sensitivity of CSEs on the starting points. Please see below the detailed response to this comments.

• Lorenz (1963) noted that when the state variable $x$ shows large values, the trajectory tends to change regimes. This suggests that if the starting point of the control is near the extreme value of $x$, we may be more difficult to control the state variables in the target regime. Here, we conduct 400 CSE experiments with different starting points randomly sampled in between the time unit of 50 and 150. When the state value $x$ of the starting points is positive, the CSE experiments are performed for 8,000 time steps, i.e., 1,000 data assimilation cycles. Figure 3 shows the relationship between the number of successful and failed CSE experiments and the state variable $x$ when the initial control is activated. If the control is activated at state variable $x$ in the range of 15–20, the failed probability is quite high. For the successful controls, the initial controls are occurred in the range from 0 to 15.

[Figure]

Figure 3: The numbers of successful and failed CSE experiments with the state variable $x$ when the initial control is activated.

• We added the discussions on the different performances of BV and SV in Figures 3b and 3c (of first submission manuscript) as follows:

"Figure 4 shows the vector changes without and under control at the initial stage of experiments. After discarding the first 8,000 time steps, we start the CSE, the control is activated at the time of 32.89. Figure 4 (a) shows the time to start control and the different trajectories without and under control by orange lines $x_n$, and magenta lines $x_c$, respectively. The empty circles are the observations generated by the CSE, which are all located in the target regime. Figures 4 (b) and 4 (c) show changes in BV and SV for experiment without and with control inputs. The lengths of the vectors represent the growth rates of BVs and SVs, which are enlarged by a factor of 20 and 50 for better visualization, respectively. When the control

inputs are added to the model, both the direction and magnitude of BV are changed due to that the external forces would change the nonlinear error propagations for $T_r$ time steps, i.e., 8 time steps. The directions and magnitudes of SV are similar for those without and under control (Fig. 4 (c)). This is because that both the NRs without and under control during the selected period do not show the trend to change regimes. Therefore, even though the control inputs are added to the NR under control, the direction and magnitude are similar to that of NR without control."

[Figure]

Figure 4: Experimental results of changes in two vectors at the initial stage of CSE. (a) State $x$ at the initial stage of experiments without and under control. The empty circles represent the observations generated by CSE. (b) The BVs. The lengths of the vectors represent the growth rate of BV with a magnification of 20. (c) The SVs. The lengths of the vectors represent the growth rate of SV with a magnification of 50. Blue line represents the spin-up states. Orange and magenta lines represent the NR without and under control, respectively. Control is activated at time 32.89.

The revisions were made in Section 3.2, Lines 148–167, Figures 3 and 4.

**Reviewer comments 1.4 (c) Results** — Lines 140–146 (Table 1): The total numbers of hit, miss, and false alarm events are not the same for BV (412) and SV (387). Were the experiments executed on the same trajectory? I suggest to verify the forecast with different time series of L63 and then computing the confidence interval of Hits, Misses, False Alarms, and threat Scores.

**Reply**: The forecast verification includes "hit", "miss", "false alarm", and "correct rejection". The "correct rejection" means that the regime change is neither observed nor forecasted by the rule. We did not show the "correct rejection" in the first submission, thus the total numbers of hit, miss and false alarm events are not the same for BV and SV.

We thank the suggestions from the reviewer to verify the forecast by the confidence interval on different time series of Lorenz-63 model. The "correct rejection" was included in the revised manuscript, accordingly, the contents and Table 1 were revised as follows:

"Table 1 shows the forecast verifications (Jolliffe & Stephenson, 2011) based on the rules of regime changes prediction described above, where "hit" means the rule successfully forecasts the observation; "miss" means that regime change is observed but not forecasted by the rule; and "false alarm" means that the rule forecasts a regime change but no regime change

occurs; "correct rejection" means that the regime change is neither observed nor forecasted by the rule. We verify the forecast for 100 different time series with each verification period equal to 1,000 time units, i.e., 100,000 time steps. The percentages of mean values and the standard deviations are shown in Table 1."

The revisions were made in Lines 177–182 and Table 1.

Table 1: Regime change forecast verification of BV and SV. The numbers and those in the bracket represent the percentage of mean and standard deviation to the averaged total number of 100 different time series.

|    | Hits (%)    | Misses (%) | False alarms (%) | Correct rejections (%) |
|----|-------------|------------|------------------|------------------------|
| BV | 33.3 (1.0)  | 9.8 (0.7)  | 5.1 (0.6)        | 51.8 (1.3)             |
| SV | 42.1 (1.4)  | 1.0 (0.5)  | 7.7 (0.7)        | 49.2 (1.7)             |

**Reviewer comments 1.4 (d) Results** — Line 148: "Fig.4d" → "Fig.4c"

**Reply**: Fixed.

The revision was made in Line 186.

**Reviewer comments 1.4 (e) Results** — Line 150: "Fig.4e" → "Fig.4d", 0.0312 does not appear on the color bar.

**Reply**: Fixed.

The revisions were made in Line 192 and Fig. 5.

**Reviewer comments 1.5 Discussion** — Line 168: "2.96" → "0.0296". It is not convinced that the maximum growth rate of SV (0.0296) is fixed for any starting point in CSEs. The authors could plot the maximum growth rate of SV as a function of the starting point to see whether the maximum growth rate varies or not.

**Reply**: Fixed: "2.96"→"0.0296".

We added the maximum growth rate of SV as a function of the starting point as follows:

"To illustrate the robustness of the reduction of the maximum growth rate of SV, we examined the SV of the successful CSEs with different starting points. The relationship between the state variable $x$ when the initial control is activated and the maximum growth rate of SV is shown in Fig. 6. The maximum growth rates of SV are affected by the starting point. Compared with the cases without control, the maximum growth rate of SV under control presents a great reduction for all CSEs with different starting points. The maximum growth rates of SV under control shows a mean value of 0.0296 and a standard deviation of 0.00015. We use the mean value of 0.0296 for our subsequent studies."

The revisions were made in Line 216, Lines 193–198 and Fig. 6.

**Reviewer comments 1.6** — For figures, please increase the line width for better visualization.

[Figure]

Figure 6: Maximum growth rate of SV for the trajectories under control at different starting points.

**Reply**: Fixed. We increased the line width of the figures.

The revisions were made in all necessary figures.
* * *
**Reviewer #2**

**General comments** — Control simulation experiment (CSE) is a method recently introduced by Miyoshi and collaborators with the aim of controlling chaotic dynamical systems. The ultimate goal is to apply this approach to the weather system, but so far only preliminary investigations have been performed on the Lorentz-63 and Lorentz-96 models. In the original paper of Miyoshi and Sun on the Lorentz-63 model, the magnitude D of the perturbations is kept constant during each experiment, and the efficiency of the control is tested for various values for D. The current manuscript discusses the use of the bred vector (BV) and of the singular vector (SV) for reducing the time of control and the total magnitude of the perturbations. More precisely, after a thorough study of the behavior of the BV and of the SV for the system under control, the authors propose to update the magnitude of the perturbations based on the maximum growth rate of the SV. On average, this implementation leads to a reduction of about 40% of the time of control, and of about 20% of the magnitude of total perturbation applied to the system.

The manuscript is clear and easy to read. In addition, since reducing the control time and the magnitude of the perturbations are two of the key factors for a successful implementation of a CSE to more complex systems, discussing any method for achieving this goal is valuable. As a consequence, the referee recommends this manuscript for publication in Nonlinear Processes in Geophysics, once the comments mentioned below are taken into account.

**Reply**: We thank the reviewer for explaining the ultimate gaol and key components of CSE, and support of our work. We are glad that our text properly highlights the improvement from

the original paper of Miyoshi and Sun (2022), which we also consider to be our take-home message.

Please see below, in blue, for a point-by-point response to the reviewers' comments and concerns. All page numbers refer to the annotated manuscript with tracked changes.

**Reviewer comments 2.1** — The first part of the investigation consists in studying the evolution of the BV and of the SV for the system under control. Indeed, it is known that these vectors can be used to predict the regime change for the Lorentz-63 model, and the aim of the CSE is precisely to keep the system always in the same regime. Quite surprisingly, Figure 4 shows a very different behavior of the BV and SV for the system under control, with large values taken by the BV despite the absence of change of regime. A qualitative difference is already visible in Figure 3. How can one understand the behavior of the BV? Is it due to an inadequate definition of the BV for the controlled system? Unfortunately, the paragraph describing the computation of the BV is not clear (lines 95 to 99). It would be useful to clarify this paragraph, and to interpret the unexpected behavior of the BV for the system under control. Along the same line, would it be possible to discuss the observation reported at the end of Section 3.2: "The directions and magnitudes of SV are similar for those without and under control", could this be expected?

**Reply**: We added an illustration on the calculation of BV for the controlled system, as shown in Fig. 1 (b). The definition of BV was revised as follows:

"For the trajectory under control $x_c$, the influence of control inputs in CSE accumulates over multiple time steps. To examine the influence of the perturbation $(\delta \frac{p}{||p||} + r)$, on the systems without and under control, we add the same control inputs (the purple arrows in Fig. 1 (b)) to the perturbed run as the independent run $x_c$ for calculating the BV."

Regarding the interpretation of BV, Evans et al. (2004) reported that BV could be used to predict the regime change for the Lorenz-63 model without control. If we control the variables of Lorenz-63 model to stay in the target regime, external forces are necessary (Miyoshi & Sun, 2022). These external forces would change the characteristics of nonlinear error propagations, e.g., BVs. We added the discussions on BV at the initial stage and for a long time period in the revised manuscript.

Figure 4 (b) shows the BVs of NRs without and under control at the initial stage of control experiment. The lengths of the vectors are the growth rates of BVs, which are enlarged by a factor of 20 for better visualization. When the control inputs are added to the model, both the direction and magnitude of BV are changed due to that the external forces would change the nonlinear error propagations for 8 time steps, which is a common definition of BV (Kalnay, 2003).

Figures 5 (a) and 5 (c) present the growth rates of BVs of NRs without and under control for a long time period. Results suggest that the growth rate of BV of controlled system shows large values despite the absence of regime changes, which gives us two hints. One is that when external forces are added to Lorenz-63 model, BV may not be able to effectively predict the regime changes. Another one is that the nonlinear error growth of the controlled system is larger than that of the original system, indicating that CSE might increase the

nonlinear features to the chaotic models. To investigate the Lyapunov dimensions through covariant Lyapunov vectors (Ginelli et al., 2007; Ginelli et al., 2013; Norwood et al., 2013) is our future study for extensively exploring the chaotic characteristics.

Regarding the Fig. 4 (c), the direction and magnitude of SV are similar for the NRs without and under control. This is because that both the NRs without and under control do not show the trend to change regimes during the selected period. Therefore, even though the control inputs are added to the NR under control, the direction and magnitude are similar to that of NR without control.

[Figure]

Figure 1: Methods for calculating (a) BV without control, (b) BV under control, and (c) SV in general case. Please refer to the text for the meaning of each parameter.

The revisions were made in Lines 98–101, Lines 157–167, Lines 185–191, and Fig. 1.

**Reviewer comments 2.2** — The BV and the SV can be computed for any dynamical system, but they turned out to be useful for predicting the change of regime for the Lorenz-63. This property is then used for rescaling the perturbation amplitude based on the knowledge of the SV. If other chaotic systems are considered, without such a clear relation between the property under control and the BV or the SV, can one still expect any use of these vectors for reducing the time of control or the magnitude of the perturbations? In other words and even if the question is very vague, is there an abstract applicability of your approach, or does it intrinsically depend on the existing link between the growth rates of the BV or SV and the change of regime?

**Reply**: We thank the reviewer for broaden our understanding to employ the proposed method to reduce the manipulations for general cases. The proposed approach might be able to reduce the manipulations of CSE for the general dynamical systems. We added the discussions in the Conclusions section.

"The information of bred vector and singular vector may be possible to be applied to reduce the time of control and magnitude of perturbations for the general cases, include but not limit to Lorenz-63 model and regime changes. The information of bred vector and singular vector may be possible to be applied to reduce the time of control and magnitude of perturbations for the general cases, include but not limit to Lorenz-63 model and regime changes. Control simulation experiment was designed to control the weather which shows separatrices, which could be defined as the separations between a normal precipitation and a heavy rainstorm, or the separations of typhoon moving east and moving west etc. The error growth near these separatrices might presents different characteristics compared with the that far away from

the separatrices, e.g., a great change of the growth rate of singular vector. Based on the information provided by the instability vectors, we may be able to find an effective way for reducing the manipulations in the control simulation experiment."

The revisions were made in Lines 248–254.

**Reviewer comments 2.3** — In Section 2.4, if To is 1 time step, does it mean that the product of matrices in (8) reduces to only 1 matrix? Similarly, if n=3, is it necessary to mention the general case for S as a n times n diagonal matrix? This short section can probably be simplified a little bit.

**Reply**: When $T_o$ is 1 time step, the product of the matrices in (8) exactly reduces to only 1 matrix. If $n = 3$, we can simplify the **S** to be only three items. We thank the suggestions from reviewer to make the contents concise and easy to read, accordingly, we revised as follows:

"For simplicity, we set $T_o$ equals to 1 time step, i.e., 0.01 time units. The SV can be calculated through the singular value decomposition of Jacobian matrix (7) as:

$$\mathbf{J}(t) = \mathbf{USV^T},$$

where **U** and **V** are orthonormal matrices (Press et al., 1992), $\mathbf{V^T}$ denotes the conjugate transpose of **V**, $\mathbf{S} = \text{diag}(s_{11}, s_{22}, s_{33})$ is a diagonal matrix with descending non-negative singular values. We calculate the first column of **V**, which is the initial leading SV, corresponding to the fastest growing vector from time $t$ to $t + 1$. Euclidean norm is employed to compute the growth rate of SV, which is given by $\ln s_{11}$.

We show the general case of calculating SV for dynamic models in Fig. 1 (c). The singular value decomposition is conducted in the tangent linear model (TLM) by the production of Jacobian matrices. In this study, the Jacobian matrices for both trajectories without and under control ($x_n$ and $x_c$) are calculated by Eq. 7, assuming that the manipulations applied in the CSE can be regarded as an external force that would not affect the linear propagation of perturbations."

The revisions were made in Section 2.4, Lines 119–136.

**Reviewer comments 2.4** — On line 168, is it really 2.96 or 0.0296?

**Reply**: Fixed.

The revisions were made in Line 216.

**Reviewer comments 2.5** — On line 203 – 204, the sentence starting with Nevertheless... should be improved.

**Reply**: We revised this sentence to be: "present study throws lights on the introduction of instability vectors to find optimal manipulations for the control simulation experiment."

The revisions were made in Lines 259–261.

**References**

Bishop, C. H., Etherton, B. J., & Majumdar, S. J. (2001). Adaptive sampling with the ensemble transform Kalman filter. Part I: Theoretical aspects. *Monthly Weather Review*, *129*(3), 420–436. https://doi.org/10.1175/1520-0493(2001)129⟨0420:ASWTET⟩2.0.CO;2

Egolf, D. A., Melnikov, I. V., Pesch, W., & Ecke, R. E. (2000). Mechanisms of extensive spatiotemporal chaos in Rayleigh–Bénard convection. *Nature*, *404*(6779), 733–736. https://doi.org/10.1038/35008013

Evans, E., Bhatti, N., Kinney, J., Pann, L., Peña, M., Yang, S.-C., & Kalnay, E. (2004). RISE: Undergraduates find that regime changes in Lorenz's model are predictable. *Bulletin of The American Meteorological Society - BULL AMER METEOROL SOC*, *85*. https://doi.org/10.1175/BAMS-85-4-520

Ginelli, F., Poggi, P., Turchi, A., Chaté, H., Livi, R., & Politi, A. (2007). Characterizing dynamics with covariant Lyapunov vectors. *Phys. Rev. Lett.*, *99*, 130601. https://doi.org/10.1103/PhysRevLett.99.130601

Ginelli, F., Chaté, H., Livi, R., & Politi, A. (2013). Covariant Lyapunov vectors. *Journal of Physics A: Mathematical and Theoretical*, *46*(25), 254005. https://doi.org/10.1088/1751-8113/46/25/254005

Houtekamer, P. L., & Zhang, F. (2016). Review of the ensemble Kalman filter for atmospheric data assimilation. *Monthly Weather Review*, *144*(12), 4489–4532. https://doi.org/10.1175/MWR-D-15-0440.1

Jolliffe, I. T., & Stephenson, D. B. (2011). Forecast verification: A practitioner's guide in atmospheric science, second edition. In *Forecast verification*. John Wiley & Sons, Ltd. https://doi.org/https://doi.org/10.1002/9781119960003

Kalnay, E. (2003). *Atmospheric modeling, data assimilation and predictability*. Cambridge University Press.

Kalnay, E., Li, H., Miyoshi, T., Yang, S.-C., & Ballabrera-Poy, J. (2007). 4-D-Var or ensemble Kalman filter? *Tellus A: Dynamic Meteorology and Oceanography*, *59*(5), 758–773. https://doi.org/10.1111/j.1600-0870.2007.00261.x

Lorenz, E. N. (1963). Deterministic nonperiodic flow. *Journal of Atmospheric Sciences*, *20*(2), 130–141. https://doi.org/10.1175/1520-0469(1963)020⟨0130:DNF⟩2.0.CO;2

Lorenz, E. N. Predictability: A problem partly solved. In: In *Seminar on predictability*. *1*. ECMWF. 1996, 1–18.

Miyoshi, T., & Sun, Q. (2022). Control simulation experiment with Lorenz's butterfly attractor. *Nonlinear Processes in Geophysics*, *29*(1), 133–139. https://doi.org/10.5194/npg-29-133-2022

Norwood, A., Kalnay, E., Ide, K., Yang, S.-C., & Wolfe, C. (2013). Lyapunov, singular and bred vectors in a multi-scale system: An empirical exploration of vectors related to instabilities. *Journal of Physics A: Mathematical and Theoretical*, *46*(25), 254021. https://doi.org/10.1088/1751-8113/46/25/254021

Press, W. H., Teukolsky, S. A., Vetterling, W. T., & Flannery, B. P. (1992). *Numerical recipes in FORTRAN. The art of scientific computing*.

Sun, Q., Miyoshi, T., & Richard, S. (2022). Control simulation experiments of extreme events with the Lorenz-96 model. *Nonlinear Processes in Geophysics Discussions*, *2022*, 1–18. https://doi.org/10.5194/npg-2022-12

Tokuda, K., Fujiwara, N., Sudo, A., & Katori, Y. (2021). Chaos may enhance expressivity in cerebellar granular layer. *Neural Networks*, *136*, 72–86. https://doi.org/https://doi.org/10.1016/j.neunet.2020.12.020

Tokuda, K., Katori, Y., & Aihara, K. (2019). Chaotic dynamics as a mechanism of rapid transition of hippocampal local field activity between theta and non-theta states. *Chaos: An Interdisciplinary Journal of Nonlinear Science*, *29*(11), 113115. https://doi.org/10.1063/1.5110327

Yang, S.-C., Kalnay, E., & Hunt, B. (2012). Handling nonlinearity in an ensemble Kalman filter: Experiments with the three-variable Lorenz model. *Monthly Weather Review*, *140*(8), 2628–2646. https://doi.org/10.1175/MWR-D-11-00313.1

---

## Author Response (AR2)

**Editorial board Member comment** — I am pleased to inform you that the two examiners and I have made our decision. We are satisfied with this new version of the manuscript and believe that the article is ready for publication. Congratulations and thank you very much for your efforts.

**Reply**: Dear Prof. Pierre Tandeo,

Thank you very much for evaluating and accepting our manuscript for publication.

We have improved some sentences and carefully reviewed the final version of manuscript based on the comments from two reviewers.
* * *
**Reviewer #1**

**Reviewer comments 1.1** — You could improve some sentences like "The error growth near these separatrices might presents different characteristics compared with the that far away from the separatrices …"

**Reply**: We revised the sentence as: " The instability vectors probably show drastic changes near these separatrices, which might give us the chance to find effective ways for reducing the manipulations in the control simulation experiment".

**Reviewer comments 1.2** — Line 18: remove "40 variables"? Not wrong, but it will be better to keep consistent with the introduction of L63.

**Reply**: Sun et al. (2022) conducted the CSE on Lorenz-96 model, which is different from Lorenz-63 model. To avoid the possible confusions, we removed "40 variables" here.

**Reviewer comments 1.3** — Lines 24-25: remove "It was demonstrated that CSE can control the trajectory in ideal models". This sentence is repeated from the previous ones.

**Reply**: Fixed.

**Reviewer comments 1.4** — Line 67: "ETKF" → "EnKF" ?

**Reply**: Fixed.

**Reviewer comments 1.5** — Line 149: "we may be" → "it may be".

**Reply**: Fixed.

**Reviewer comments 1.6** — Lines 189–191: Remove "To investigate the Lyapunov dimensions through covariant Lyapunov vectors (Norwood et al., 2013; Ginelli et al., 2013, 2007) is our future study for extensively exploring the chaotic characteristics". This sentence is inappropriate to be here as you have only done experiments with BV and SV. And you already added this perspective in Section Conclusion.

**Reply**: Fixed.

**Reviewer comments 1.7** — In general, we encourage you to carefully review your final version.

**Reply**: We sincerely thank the reviewers and editors for reading and commenting our manuscript. We have carefully checked the final version, and hope it to be published in the Nonlinear Processes in Geophysics.